# Comparative analysis of tight-binding models for transition metal dichalcogenides

**Bert Jorissen⋆, Lucian Covaci and Bart Partoens**

Department of Physics, University of Antwerp,
Groenenborgerlaan 171, B-2020 Antwerp, Belgium

⋆ bert.jorissen@uantwerpen.be

## Abstract

We provide a comprehensive analysis of the prominent tight-binding (TB) models for transition metal dichalcogenides (TMDs) available in the literature. We inspect the construction of these TB models, discuss their parameterization used and conduct a thorough comparison of their effectiveness in capturing important electronic properties. Based on these insights, we propose a novel TB model for TMDs designed for enhanced computational efficiency. Utilizing $MoS_2$ as a representative case, we explain why specific models offer a more accurate description. Our primary aim is to assist researchers in choosing the most appropriate TB model for their calculations on TMDs.

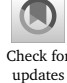

# 1   Introduction

Monolayers of transition metal dichalcogenides (TMDs) can be regarded as the semiconductor equivalent to graphene [1,2]. They are two dimensional (2D) materials with a hexagonal symmetry, similar to graphene. In the unit cell they have one metal ($M$) atom and two chalcogen atoms ($X$), written as $MX_2$ with $M = (Mo, W)$ and $X = (S, Se, Te)$. The TMDs have a direct band gap around the two inequivalent $K$ and $K'$ points at the corners of the Brillouin zone [3,4]. This gap is in the range of visible light. The spin and valley degrees of freedom are coupled what gives rise to interesting phenomena such as spin and valley dependent optical transitions [5]. This makes TMDs interesting candidates for spintronic, electronic, optoelectronic and valleytronic applications.

More insights can be gained by studying these phenomena using theoretical methods. Density functional theory (DFT) gives an *ab initio* description of these materials, but can only be used for smaller systems. Alternatively, simple two-band effective models are constructed using symmetry considerations. These accurately describe the behaviour of massive Dirac fermions seen at the band-edge [6,7]. However, these simple models are only valid close to the band-edge. To get a better understanding of the electronic structure of the material, different tight-binding (TB) models were proposed in the literature to give an improved description of the electronic structure.

Compared with graphene, the TB models for TMDs are more involved. In graphene, a simple model describing the $p_z$-orbitals with the out-of-plane $\pi$-bond already gives reasonable results, even with just nearest neighbour (NN) hoppings [8]. In TMDs on the other hand, the conduction and valence band have rich orbital contributions due to the hybridization of the $d$-orbitals from the metal atom with the two $p$-orbitals of the chalcogen atoms. This makes it harder to construct a basic model that can accurately describe the electronic structure of the system, which explains the large number of proposed TB models for TMDs.

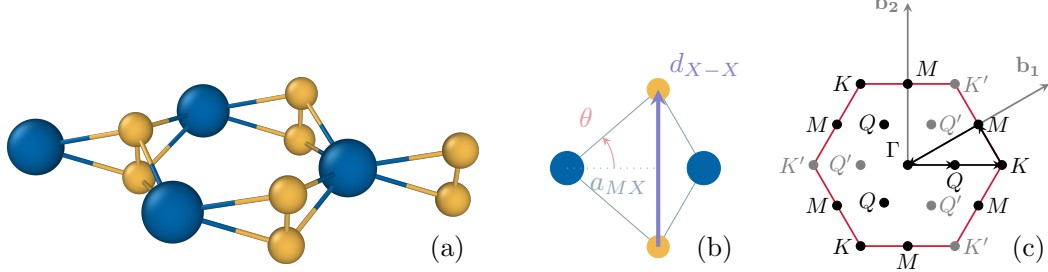

Figure 1: (a) Perspective view of the $TMD-H$ structure with the different $X^b-M-X^t$ indicated in yellow-blue-yellow respectively, (b) side view of the structure, (c) the Brillouin Zone with the high-symmetry points with emphasis on the inequivalent $K$ and $K'$ points.

This work presents an overview of the relevant TB models for TMDs in literature and proposes a new TB model. The different steps in the construction of the Hamiltonian for the TB model are discussed and how these are connected to symmetries in the TMDs. The TB models are divided in two categories: TB models that take the Slater Koster (SK) [9] two center approximations as basis and ones that take the symmetry group (SG) as a starting point when constructing the Hamiltonian. Furthermore, these TB models are divided according to the orbitals they include and the amount of hoppings they take into account. We give a clear comparison between all these TB models to indicate their strengths and weaknesses, and correspondingly their possible applications. Based on this, we propose a new SG TB model which has a good balance between accuracy and computational efficiency.

The TB models from literature were constructed with various objectives, such as getting a good approximation around certain points in the Brillouin zone, a better fit for the highest valence and lowest conduction bands or for multiple bands. To better compare these models, we refit the parameters to DFT results for $MoS_2$. In this work the models are describing the electronic properties of $MoS_2$, other materials follow the same trends due to their similar structures.

The uniform construction of the Hamiltonian allows for an elegant implementation of all the discussed TB models. They are all implemented within a new Python package *TMDybinding* [10] that builds upon the Pybinding software [11]. This package simplifies the construction of the Hamiltonian, independent from the TB model that is used in the calculation.

The paper is further organized as follows. In section 2 we present the construction of the Hamiltonian for the TB models considering the relevant orbitals, symmetries, hopping and spin-orbit coupling (SOC) terms. The construction procedures and properties of these TB models are discussed and the new SG model is introduced in this section. We then explain the methods used to fit the parameters in the TB models to the DFT results in section 3 and we compare these fitted models against the DFT results considering different properties: the DOS and its orbital contribution, the band structure, the effect of SOC and the Berry phase. Finally, we give a summary of the models and identify which model to use for specific situations and the possible applications in large-scale systems in section 4.

## 2 Tight-binding models for TMDs

### 2.1 Space group and symmetries

A single $TMD-H$ layer has a hexagonal structure as shown in figure 1. The unit cell consists of a metal atom ($M$) and two out-of-plane chalcogen atoms ($X$). For MoS$_2$, the metal and chalcogen atoms are $Mo$ and $S$ respectively. The in-plane distance from the metal atom to the chalcogen atom is $a_{MX}$, the distance between bottom ($X^b$) and top ($X^t$) the chalcogen atoms is $d_{X-X}$ and the angle between the plane of metal atoms and the line connecting the metal and chalcogen atom is $\theta$. The lattice vectors are given by $\mathbf{a_1} = a\hat{x}$ and $\mathbf{a_2} = a\left(-\frac{1}{2}\hat{x} + \frac{\sqrt{3}}{2}\hat{y}\right)$, with $a = \sqrt{3}a_{MX}$. For MoS$_2$, we obtained the values $a = 3.19$Å, $d_{X-X} = 3.12$Å and $\theta = 0.703$ using density function theory (DFT) calculations. The reciprocal-space vectors are $\mathbf{b_1} = \frac{2\pi}{a}\left(\hat{x} + \frac{1}{\sqrt{3}}\hat{y}\right)$ and $\mathbf{b_2} = \frac{4\pi}{a\sqrt{3}}\hat{y}$. The special points in the first Brillouin zone (BZ) are at $\Gamma = \mathbf{0}$ (the center), $K = -K' = \frac{1}{3}(2\mathbf{b_1} - \mathbf{b_2})$ (the corners), and $M = \frac{1}{2}\mathbf{b_1}$ (the center of the edges). The line connecting these points is a line of high-symmetry. Around the $Q$ point, between the $\Gamma$ and $K$ points, the conduction band has a local minimum.

The outer orbitals of the TMD are the $d$- and $p$-orbitals for the metal and chalcogen atom respectively, which makes a total of eleven orbitals

$$
\begin{aligned}
&\left(d_{z^2}, d_{xz}, d_{yz}, d_{x^2-y^2}, d_{xy}\right), \\
&\left(p_x^t, p_y^t, p_z^t\right), \quad \left(p_x^b, p_y^b, p_z^b\right),
\end{aligned}
\tag{1}
$$

where $p^t$ and $p^b$ denote the orbitals from the top and bottom chalcogen respectively.

The $TMD-H$ structure belongs to the $D_{3h}$ group and has a mirror symmetry in the $xy$-plane $\hat{\sigma}_h$, three mirror symmetries $\hat{\sigma}_v$ along the armchair direction, two threefold rotation axes $\hat{C}_3$ through each atomic position, three twofold rotation axes $\hat{C}_2'$ along the armchair direction and two $\hat{S}_3$-symmetries. The irreducible representations for the orbitals in the Mulliken notation is $A_1'\left(d_{z^2}\right)$, $A_2''\left(p_z\right)$, $E'\left(p_x, p_y, d_{x^2-y^2}, d_{xy}\right)$ and $E''\left(d_{xz}, d_{yz}\right)$. Under the $\hat{\sigma}_h$-symmetry, the $p$-orbitals transform into each other. These $p$-orbitals are combined to give new orbitals that are even ($e$) or odd ($o$) under this symmetry, given by [12–14]

$$
\begin{aligned}
p_{x/y}^e &= \frac{1}{\sqrt{2}}\left(p_{x/y}^t + p_{x/y}^b\right), & p_z^e &= \frac{1}{\sqrt{2}}\left(p_z^t - p_z^b\right), \\
p_{x/y}^o &= \frac{1}{\sqrt{2}}\left(p_{x/y}^t - p_{x/y}^b\right), & p_z^o &= \frac{1}{\sqrt{2}}\left(p_z^t + p_z^b\right).
\end{aligned}
\tag{2}
$$

With these new $p$-orbitals, the atomic basis is divided into four parts: the parts that are even and odd under $\hat{\sigma}_h$ and the parts given by the the metal ($M$) and chalcogen ($X$) atoms,

$$
\phi = \left(\phi^{M,e}, \phi^{X,e}, \phi^{M,o}, \phi^{X,o}\right),
\tag{3}
$$

with

$$
\begin{aligned}
\phi^{M,e} &= \left(d_{z^2}^e, d_{x^2-y^2}^e, d_{xy}^e\right), & \phi^{X,e} &= \left(p_x^e, p_y^e, p_z^e\right), \\
\phi^{M,o} &= \left(d_{xz}^o, d_{yz}^o\right), & \phi^{X,o} &= \left(p_x^o, p_y^o, p_z^o\right).
\end{aligned}
\tag{4}
$$

It is assumed that this atomic basis is orthonormal, there is no overlap between the different orbitals.

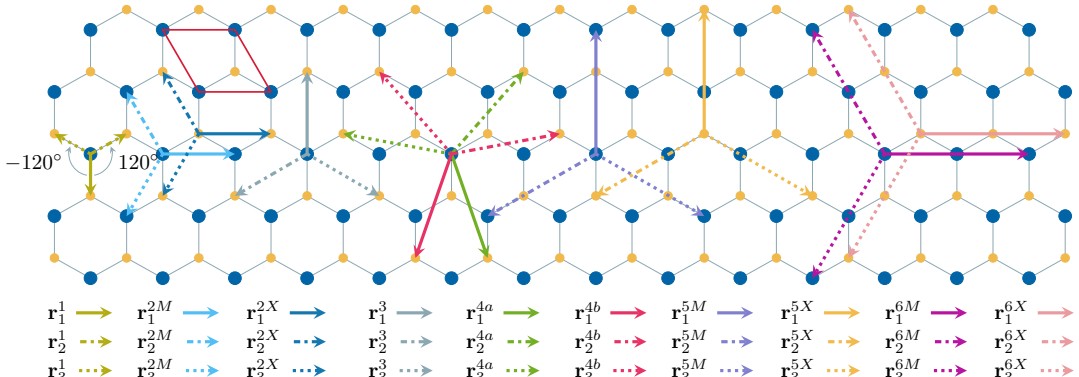

**Figure 2:** The hexagonal structure of the TMD with the metal atoms at the blue larger circles and the chalcogen atoms at the smaller yellow circles. The unit cell is indicated in the upper left corner in red. The in-plane hopping vectors $\mathbf{r}_l^\delta$ are grouped per $n$-th NN from left to right. For each of these groups, the hopping matrix $T_{1,\delta}^{\mu\nu,\rho}$ associated with $\mathbf{r}_1^\delta$ is given in appendix B. The hoppings 4$a$ and 4$b$ can be considered as one hopping along the armchair direction, that is rotated along small angles.

## 2.2 Tight-binding Hamiltonian

Using the atomic basis from (3), we write down a general description for the tight-binding Hamiltonian as

$$H_{TB} = \langle \phi | \hat{H} | \phi \rangle = \begin{pmatrix} H_{TB}^e & 0 \\ 0 & H_{TB}^o \end{pmatrix}, \tag{5}$$

with

$$H_{TB}^\rho = \begin{pmatrix} \varepsilon^{M,\rho} & 0 \\ 0 & \varepsilon^{X,\rho} \end{pmatrix} + \begin{pmatrix} T^{MM,\rho} + \left(T^{MM,\rho}\right)^\dagger & T^{XM,\rho} \\ \left(T^{XM,\rho}\right)^\dagger & T^{XX,\rho} + \left(T^{XX,\rho}\right)^\dagger \end{pmatrix}, \tag{6}$$

and $\rho = \{e, o\}$, the Hamiltonian is block diagonal in the even and odd parts under $\hat{\sigma}_h$. The onsite and hopping energies are indicated as $\varepsilon^{\mu,\rho}$ and $T^{\mu\nu,\rho}$, with $\{\mu, \nu\} = \{M, X\}$.

## 2.3 Hopping energies and rotations

The hopping energies $T^{\mu\nu,\rho}$ connect the different orbitals. For the TB models with multiple orbitals considered here, these hopping energies are represented by matrices. The values of these hopping matrices are different for the different TB models.

In general, they consist of several different $n$-th nearest neighbour (NN) hopping matrices between the metals and chalcogen atoms. Written in reciprocal space, these matrices also have an additional phase dependence on the distance between the orbitals,

$$T^{\mu\nu,\rho} = \sum_\delta \sum_{l=(1,2,3)} T_{l,\delta}^{\mu\nu,\rho} e^{i\mathbf{k}\cdot\mathbf{r}_l^\delta}, \tag{7}$$

with $l$ the direction of this hopping and $\delta$ denoting the $n$-th NN-hopping as given in figure 2,

$$\delta = \begin{cases} 1, 3, 4a, 4b, & \text{when} \quad \mu\nu = XM, \\ 2\mu, 5\mu, 6\mu, & \text{when} \quad \mu = \nu. \end{cases}$$

Depending on the TB model, a set of different hoppings $\delta$ is taken, like $\delta \in \{1, 2M, 2X\}$. The furthest hopping from this set is $\delta_{max}$.

The three matrices associated with $l \in \{1, 2, 3\}$ per hopping $\delta$ are related by the $\hat{C}_3$-symmetry. Only one direction for each hopping $\delta$ from $\nu$ to $\mu$ must be specified, the other two are obtained from $l = 1$ as

$$T_{2,\delta}^{\mu\nu,\rho} = R^{\mu,\rho} T_{1,\delta}^{\mu\nu,\rho} (R^{\nu,\rho})^T , \qquad T_{3,\delta}^{\mu\nu,\rho} = (R^{\mu,\rho})^T T_{1,\delta}^{\mu\nu,\rho} R^{\nu,\rho} , \qquad (8)$$

with the rotation matrices given by

$$R^{X,\rho} = \begin{pmatrix} \cos\gamma & -\sin\gamma & 0 \\ \sin\gamma & \cos\gamma & 0 \\ 0 & 0 & 1 \end{pmatrix}, \qquad R^{M,e} = \begin{pmatrix} 1 & 0 & 0 \\ 0 & \cos 2\gamma & -\sin 2\gamma \\ 0 & \sin 2\gamma & \cos 2\gamma \end{pmatrix},$$
$$R^{M,o} = \begin{pmatrix} \cos\gamma & -\sin\gamma \\ \sin\gamma \cos\gamma \end{pmatrix}, \qquad (9)$$

with $\gamma = 120°$. The matrix $R^{M,e}$ transforms the orbitals over double the rotation angle $\gamma$ because of the $|m_l| = 2$ azimuthal quantum number for the $(d_{x^2-y^2}, d_{xy})$-orbitals.

For example, the even ($\rho = e$) first nearest neighbour hoppings ($\delta = 1$) from the metal atom ($\nu = M$) to the chalcogen atoms ($\mu = X$) are given by the three vectors $\mathbf{r}_l^1$, where the three matrices given by $l \in \{1, 2, 3\}$ are related by

$$T_{2,1}^{XM,e} = R^{X,e} T_{1,1}^{XM,e} (R^{M,e})^T , \qquad T_{3,1}^{XM,e} = (R^{X,e})^T T_{1,1}^{XM,e} R^{M,e} . \qquad (10)$$

## 2.4 Hopping energies and symmetries

The exact formulation of the hopping matrices depends on the orbitals that are included in the TB model $\phi$ and the set of nearest neighbour hoppings $\delta$. All the symmetries present in the symmetry group for the TMD must be considered by the TB model and its hopping matrices. The $\hat{\sigma}_h$-symmetry is already satisfied with the even ($e$) and odd ($o$) formulation of the orbitals and the $\hat{C}_3$-symmetry by rotating the hopping matrices with the expressions in equation (8).

The other important symmetry operation is the $\hat{\sigma}_v$ symmetry. This last symmetry will further reduce the amount of parameters to a minimal set, giving the final shape of the hopping matrices. Here, we will consider the mirror plane through the $yz$-axes ($\hat{\sigma}_v^1$). The two other mirror planes can be considered by a rotation as given by the matrices in equation (9). The basis $\phi$ splits in even ($g$) and odd ($u$) parts under $\hat{\sigma}_v^1$, given by the orbitals

$$\phi^g = \left(d_{z^2}, d_{x^2-y^2}, d_{yz}, p_z^e, p_y^e, p_z^o, p_y^o\right), \qquad \phi^u = \left(d_{xy}, d_{xz}, p_x^e, p_x^e\right) . \qquad (11)$$

When the Hamiltonian is written in this basis, it has parts that are even and odd under $\hat{\sigma}_v^1$. Here, we will only write down the hopping matrix for a general hopping between the even ($g$) and odd ($u$) sectors,

$$T_{1,\delta}^{\mu\nu,\rho} = \begin{pmatrix} T_{gg} & T_{gu} \\ T_{ug} & T_{uu} \end{pmatrix}, \qquad (12)$$

with $T_{gg}$ and $T_{uu}$ denoting the hoppings between orbitals that are even and odd with $\hat{\sigma}_v^1$ respectively, $T_{gu}$ and $T_{ug}$ denote the hoppings that take a hopping from an even to an odd sector and vice versa.

The hoppings along the armchair directions ($\delta \in \{1, 3, 4a, 4b, 5M, 5X\}$) are parallel to the mirror plane of the $\hat{\sigma}_v^1$-symmetry. Under this $\hat{\sigma}_v^1$ symmetry transformation, the matrix is

invariant

$$
\begin{aligned}
T &= \mathcal{D}[\hat{\sigma}_v^1] T \mathcal{D}[\hat{\sigma}_v^1]^\dagger \\
&= \begin{pmatrix} T_{gg} & (-1)T_{gu} \\ (-1)T_{ug} & (-1)(-1)T_{uu} \end{pmatrix} \\
&= \begin{pmatrix} T_{gg} & 0 \\ 0 & T_{uu} \end{pmatrix},
\end{aligned} \tag{13}
$$

the sections that connect the even and odd parts will disappear for hoppings along the armchair direction.

For the hoppings along the zigzag directions ($\delta \in \{2M, 2X, 6M, 6X\}$) the mirror plane will be perpendicular to the hopping. The symmetry thus flips the hopping from going to the right to going to the left giving the Hermitian conjugate of the hopping,

$$
\begin{aligned}
T^\dagger &= \begin{pmatrix} T_{gg}^\dagger & T_{ug}^\dagger \\ T_{gu}^\dagger & T_{uu}^\dagger \end{pmatrix} \\
&= \mathcal{D}[\hat{\sigma}_v^1] T \mathcal{D}[\hat{\sigma}_v^1]^\dagger \\
&= \begin{pmatrix} T_{gg} & (-1)T_{gu} \\ (-1)T_{ug} & (-1)(-1)T_{uu} \end{pmatrix} \\
&= \begin{pmatrix} T_{gg} & -T_{gu} \\ T_{gu}^\dagger & T_{uu} \end{pmatrix},
\end{aligned} \tag{14}
$$

so that $T_{gg} = T_{gg}^\dagger$, $T_{uu} = T_{uu}^\dagger$ and $T_{gu} = -T_{ug}^\dagger$.

The remaining $\hat{C}_2'$- and $\hat{S}_3$-symmetries are a combination of $\hat{\sigma}_h$-, $\hat{\sigma}_v$- and/or $\hat{C}_3$-symmetries that are already satisfied by the hopping matrices. For example, $\hat{C}_2'$ can be obtained by combining the two mirror symmetries $\hat{\sigma}_h$ and $\hat{\sigma}_v$. Therefore, a TB model constructed based on the rules related to the $\hat{\sigma}_h$ (even/odd), $\hat{C}_3$ (rotations) and $\hat{\sigma}_v$ (shape of matrix) symmetries agrees with all the symmetries in the $D_{3h}$ symmetry group of the TMD.

The hopping matrices will have different shapes depending on the hopping $\delta$ considered and the orbitals $\phi$ that are included in the TB model. The values for the parameters in these hopping matrices depend on the used TB model. We call TB models that use all the parameters allowed by the symmetry group (SG), the SG TB models. The matrices $\varepsilon^{\mu,\rho}$ and $T_{1,\delta}^{\mu\nu,\rho}$ for the onsite and hopping energies respectively, are given in appendices A and B.

## 2.5 Broken rotation symmetry

When constructing the Hamiltonian, care has to be taken to determine if the hopping is taken with the correct angle and direction. When the hopping is calculated along the wrong angle, say a rotation of $180°$ ($-\mathbf{r}_l^\delta$) from the correct hopping ($\mathbf{r}_l^\delta$), this corresponds to applying a $\hat{C}_2$-symmetry. This $\hat{C}_2$ rotation can be expressed as a combination of $\hat{\sigma}_v$ and $\hat{\sigma}_d$. The $\hat{\sigma}_d$-symmetry is a mirror symmetry along the zigzag direction. This $\hat{\sigma}_d$-symmetry is not a member of the symmetry group for the TMDs as the positions of the metal and chalcogen atoms are not interchangeable. The $\hat{C}_2$ is thus not in the symmetry group.

It is possible to transform the TB model to a different basis, connected with a $\hat{\sigma}_d$ operation. Some models in literature [12, 15] use this different basis, where the position of the metal and chalcogen atoms are swapped. The parameters given in appendix B and the *TMDybinding* package always use the same convention for the position of the metal and chalcogen atoms.

## 2.6 Slater-Koster parameterization

The hopping matrices formed by the SG approach still contain a considerable amount of parameters. The two-center integral approximation from Slater and Koster (SK) [9] gives a simplified generalized description for these parameters. Here, only hoppings between pure, hydrogen-like, non-hybridized $p$- or $d$-orbitals and bond types between the orbitals are considered. There are three different types of bonds for the TMDs considered here, $\sigma$- or $\pi$-bonds between $p$- and/or $d$-orbitals and a $\delta$-bond between the $d$-orbitals. For example, a hopping matrix between the even $d$-orbitals at different sites can have up to 6 different parameters according to the SG approach. However, with the SK approach, this reduces to only 3 parameters, $V_{dd\sigma}$, $V_{dd\pi}$ and $V_{dd\delta}$. These parameters are the same for hoppings $\mathbf{r}_l^{\delta}$ with the same radial distance $|\mathbf{r}_l^{\delta}|$.

In the SK approach, the hopping matrix between a metal atom and a chalcogen atom has a contribution for the hopping from the even $p_x^e$ to the even $d_{xy}$ given by

$$\left(T_{l,\delta}^{XM,e}\right)^{\dagger}_{(p_x^e, d_{xy})} = \sqrt{3}l^2 m V_{pd\sigma}^{e,\delta} + m(1-2l^2)V_{pd\pi}^{e,\delta}, \tag{15}$$

which is the element $E_{x,xy}$ from table 2 in Slater and Koster [9] with $l$, $m$ and $n$ the components of the unit vector from the $p_x^e$- to $d_{xy}$-orbital,

$$-\frac{\mathbf{r}_l^{\delta}}{|\mathbf{r}_l^{\delta}|} = \begin{pmatrix} l \\ m \\ n \end{pmatrix}. \tag{16}$$

This approach can be used for all the parameters in the hopping matrices.

The SK approach always produces hopping parameters that are allowed by the symmetry group of a specific material, but does not take into account every broken symmetry in the material. For example, it does not consider the $T_{gu}$ hopping elements from equation (14). Due to the hybridization in the TMDs, the orbitals deviate from the simple, hydrogen-like orbitals. These hybridized orbitals do break certain symmetries, allowing for more parameters in the full SG hopping matrices that are not included in the SK approach [16].

In the SK term from equation (15) the parameters are already divided in the even and odd terms under $\hat{\sigma}_h$. However, if the terms are constructed from the original top- and bottom-$p$-orbitals, there will be an additional cross hopping term between these top- and bottom-$p$-orbitals for hoppings $\delta \in \{2X, 5X, 6X\}$. This cross terms leads to additional non-zero elements, also in the $T_{gu}$-part, that allow for a better description of the TB model.

In appendix D the most general parameterization of the SK approach is noted down, including the splitting in even and odd parameters and the cross-hopping term between the top- and bottom-$p$-orbitals for the hoppings $\delta = \{2X, 5X, 6X\}$.

## 2.7 Spin-orbit coupling

The spin-orbit coupling (SOC) in TMDs leads to strong modifications in the band structure [7]. To include this effect, the Hamiltonian is expanded in a spin-up ($\uparrow$) and spin-down ($\downarrow$) part,

$$H_{SOC} = \begin{pmatrix} H^{\uparrow\uparrow} & H^{\uparrow\downarrow} \\ H^{\downarrow\uparrow} & H^{\downarrow\downarrow} \end{pmatrix}$$

$$= \mathbb{1}_{2X2} \otimes H_{TB} + t_{SOC}. \tag{17}$$

The spin-orbit coupling is included in the models by a $\mathbf{L} \cdot \mathbf{S}$-term

$$t_{SOC} = \sum_{\mu} \lambda^{\mu} \mathbf{L}^{\mu} \cdot \mathbf{S}^{\mu}, \tag{18}$$

Table 1: Properties of the TB models. With the exception of TB models that include the $s$-orbital [17,18] and overlap matrices [18,19], all relevant models in literature can be reduced to a model in this table. The $d$- and $p$-orbitals determine if a spin flip can happen. The full $p$-orbitals from the $X$ atoms must be present to allow the model to be transformed into one with separate orbitals for the $p$-orbitals at the $X$-top and $X$-bottom atoms. The third NN hopping for Fang is between the even orbitals.

| Model | Ref | Type | Bands | Params | Set of hoppings $\delta$ | Basis | spin flip | $X$-top-bottom |
|---|---|---|---|---|---|---|---|---|
| Rostami | [13] | SK | 6 | 13 | $\{1,2M\}$ | $\phi^{M/X,e}$ | No | No |
| Cappelluti | [20] | SK | 11 | 14 | $\{1,2M,2X\}$ | $\phi^{M/X,e/o}$ | Yes | Yes |
| Dias | [14] | SK | 11 | $29^\star$ | $\{1,2M,2X,5M,5X\}$ | $\phi^{M/X,e/o}$ | Yes | Yes |
| Liu$_2$ | [21] | SG | 3 | 8 | $\{2M\}$ | $\phi^{M,e}$ | No | No$^\dagger$ |
| Liu$_6$ | [21] | SG | 3 | 19 | $\{2M,5M,6M\}$ | $\phi^{M,e}$ | No | No$^\dagger$ |
| Wu | [22] | SG | 5 | 28 | $\{2M,5M,6M\}$ | $\phi^{M,e/o}$ | Yes | No$^\dagger$ |
| This work | | SG | 6 | 21 | $\{1,2M,2X\}$ | $\phi^{M/X,e}$ | No | No |
| Fang | [12] | SG | 11 | $41^\triangle$ | $\{1,2M,2X,3^{(e)}\}$ | $\phi^{M/X,e/o}$ | Yes | Yes |

($\star$) The original work [14] has two more unused parameters.

($\dagger$) No $X$ atoms in model.

($\triangle$) The original work [12] has one parameter less than a follow-up work [15].

with

$$\mathbf{L}^\mu \cdot \mathbf{S}^\mu = L_z^\mu S_z^\mu + \frac{1}{2}\left(L_+^\mu S_-^\mu + L_-^\mu S_+^\mu\right). \tag{19}$$

The summation goes over the different types of atoms $\mu$, with matrices that connect both the spin-up and spin-down components. The exact formulation of the SOC matrices in the expanded orbital basis of (3) is given in appendix C.

The main contribution to the SOC is given by the $L_z^\mu S_z^\mu$-term. When the other terms are neglected, the additional SOC hopping terms only couple bands with the same parity $\rho$ under $\hat{\sigma}_h$. The Hamiltonian stays block-diagonal. The broken inversion symmetry leads to a spin splitting at $K$ and $K'$. The time reversal symmetry is preserved, so that spin-up states at valley $K$ are degenerate with spin-down states at valley $K'$ and vice versa. The same splitting occurs at the $Q$ and $Q'$-points.

The other spin terms give rise to a spin-flip. This spin-flip coincides with a change in $|m_l|$ azimuthal quantum number. This SOC breaks the parity $\rho$, resulting in a denser Hamiltonian. The SOC is best described in the chiral basis, given by

$$p_0^\rho = p_z^\rho, \qquad p_{\pm 1}^\rho = \frac{1}{\sqrt{2}}\left(\mp p_x^\rho - i p_y^\rho\right),$$

$$d_0 = d_{z^2}, \qquad d_{\pm 1} = \frac{1}{\sqrt{2}}\left(\mp d_{xz} - i d_{yz}\right), \qquad d_{\pm 2} = \frac{1}{\sqrt{2}}\left(d_{x^2-y^2} \pm i d_{xy}\right). \tag{20}$$

This chiral basis is a linear combination of orbitals in the Cartesian basis with the same azimuthal $|m_l|$ and angular $l$ quantum number. The orbitals in this basis have a well defined $m_l$-quantum number, which simplifies the description of the $L_\pm^\mu$ ladder operator.

## 2.8 Description of the models

The TB models from literature are divided in two categories, models that use the SK [13,14,17, 20,23–29] or SG [12,15,21,22] approach. The different TB models have certain orbital bases $\phi^{\mu,\rho}$ and a selection of hoppings $\delta$, up to $\delta_{max}$, as summarized in table 1. The parameters used in these models were fitted against *ab initio* calculations.

The SG TB models from Liu *et al* [21] are the simplest TB models and only include the even $d$-orbitals, $\phi^{M,e}$, resulting in 3 band TB models. These $d$-orbitals are effective orbitals originating from the hybridization between the $d$- and $p$-orbitals [30]. Though they are categorised with $\delta_{max} = 2$ (Liu$_2$) or $\delta_{max} = 6$ (Liu$_6$), the models only have metal atoms, such that they are effective TB models with only $\delta_{max} = 1$ or $\delta_{max} = 3$ as maximal nearest neighbour hopping lengths in practice. An extension to the Liu$_6$ model was given by Wu *et al* [22] where they also included the odd $d$-orbitals giving a total of 5 bands, with the basis given by $\phi^{M,e/o}$.

Fang *et al* [12, 15] reported a SG model with all the orbitals $\phi^{M/X,e/o}$ up to $\delta_{max} = 3$. Compared to Liu$_6$/Wu, it also includes the deeper energy-levels in the valence band.

The models from Rostami *et al* [13], Cappelluti *et al* [20] and Dias *et al* [14] are all SK models. These models neglect the cross-hopping between $p^t$- and $p^b$-orbitals on different sites. Dias *et al* splits the model in even and odd parameters.

## 2.9 A new TB model

We present a novel SG TB model utilizing even orbitals as the basis ($\phi^{M/X,e}$) and incorporating hoppings up to second nearest neighbours. For the TMDs investigated in this work, the energy bands around the band gap are all given by contributions from the even orbitals. By only including these even orbitals, the model reduces the number of bands from 11 to 6 while preserving the important features around the band gap. The limited number of hoppings in the model ($\delta_{max} = 2$) results in fewer off-diagonal elements yielding an overall sparser matrix which is easier to solve. The chosen combination of hoppings and orbitals makes this model ideally suited for the study of larger systems. The inclusion of the chalcogen atoms in the set of even orbitals ensures a good description for calculations that need these atoms, as for example required in the study of nanoribbons.

However, the models from Fang or Liu$_6$ will provide a more accurate description, but they are computationally more demanding or do not include the chalcogen atoms, respectively. The model presented here cannot account for terms introduced by out-of-plane deformations, impurities or spin flips as these will mix the even and odd orbitals in the Hamiltonian which the new model cannot describe. The decomposition in top- and bottom $p$-orbitals is not straightforward, making it less suited for calculations on heterostructures.

In order to provide a clear comparison, we summarize the properties of our new model alongside existing models from the literature in table 1. The parameters for this new model and the existing models are given in appendices B and D. All the models are implemented in TMDybinding, as detailed in appendix H.

# 3 Calculations

## 3.1 Fitting the parameters

In the literature, each model was fitted or constructed with a different objective. This makes a direct comparison difficult. Therefore, we refitted all the models on results obtained from a DFT-calculation.

The energy bands from the DFT calculation were disentangled, i.e. the overlap of the wavefunction at neighbouring points along the path in the reciprocal space $\langle u_n^{\mathbf{k}_i} | u_m^{\mathbf{k}_{i+1}} \rangle$ was used to differentiate between a crossing and an avoided crossing. We used the function *linear sum assignment* from Scipy [31] to obtain the ordering of indices $n$, $m$ where the overlap of the wavefunction was the largest. Then, we sorted these disentangled bands according to energy at the $K$-point. We neglect the energy bands from DFT that are not taken into account by the

Table 2: Ranking of the weights in the cost function from large to small contribution to fit the parameters in the TB models to the DFT results. The bang gap (BG) energy levels are the highest and lowest energy levels in the valence and conduction band respectively. The high-symmetry path is the path that connects the $\Gamma$, $M$, $K$ and $Q$ points.

| $i$ | Property |
|---|---|
| 1 | Effective mass at the $K$ point |
| 2 | Effective mass at the $\Gamma$ point |
| 3 | BG energy levels at the $\Gamma$, $M$, $K$ and $Q$ points |
| 4 | BG energy levels along the high-symmetry path |
| 5 | All energy levels along the high-symmetry path |

TB models. The bands from the fitted TB model are also disentangled and sorted in the same manner.

The fitting procedure uses a cost function where the various objectives have different weights. The weights were selected to enhance the precision of fitting specific properties within the TB model, a summary is given in table 2. Each objective has a cost function that is based on the sum of the squared error between the function that is fitted and the objective function derived from the DFT calculation, resulting in the total cost function:

$$C = \sum_i w_i \sum_j \left( O_{i,j}^{TB} - O_{i,j}^{DFT} \right)^2 , \tag{21}$$

with $C$ the cost, $w_i$ the different weights per objective $i$ and $O_{i,j}^{TB}$, $O_{i,j}^{DFT}$ the values for the objective functions for objective $i$ and the different contributions to that objective $j$, such as the values for the energy at different levels or points in the BZ. This cost function is minimized using the *minimize* function of SciPy [31] for starting configurations given by the original parameters from the literature.

We found that the SG TB models always converged to a reasonable minimum, where a small change in weight for a specific objective leads to a small change in the parameters. However, we never obtained a parameterization for the TB models using the SK approach that fitted well to the DFT results like we found in the case of the SG approach. The SK models also had difficulties in obtaining the correct orbital character for the energy bands.

## 3.2 DFT calculations

We use the results from DFT-PBE [32] calculations in the GGA-approach as a reference for refitting the models and comparing the results obtained from TB calculations. Although more precise methods are available, the aim is to compare the models in the ability to capture the precise details from a given structure.

The DFT calculations were preformed with Abinit [33] using pseudopotentials obtained from PsueodoDojo [34]. Norm-conserving full-relativistic [33] pseudopotentials were used for the calculations with SOC and PAW [34] pseudopotentials for the spin degenerate calculations. The plane-wave energy cutoff was $40 Ha$ on a $10 \times 10 \times 1$ $\Gamma$-centered Monkhorst-Pack grid of $k$-points. The monolayers were separated with a 25Å vacuum region. The vacuum level was chosen as reference for the energy.

Abipy [35] was used to analyze the DFT wavefunction to calculate the Berry phase, spin-polarisation, band disentangling and projected DOS.

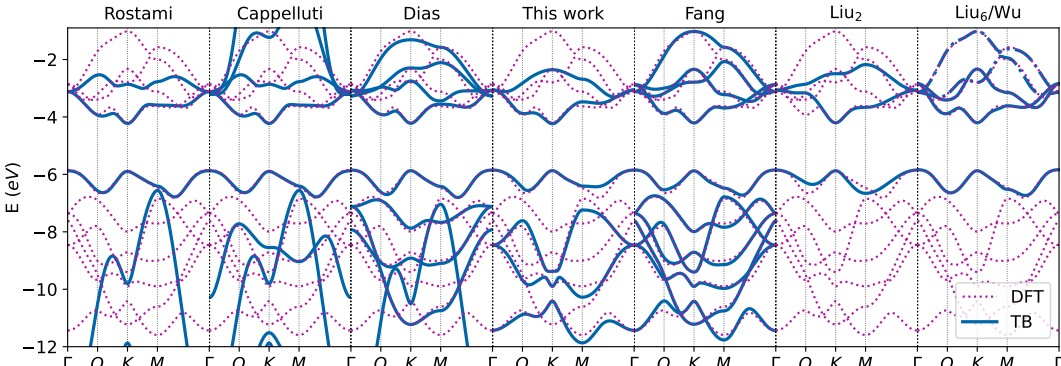

Figure 3: The band structure for the different TB models given in table 1. The DFT results on which the TB models were fitted are given as a reference. The additional odd bands in the Wu model are indicated with a dotted line.

### 3.3 TB calculations

All the models are implemented in the Python software library Pybinding [11]. We created an extension, TMDybinding, that allows the user to easily select which model they want to use and preform the calculations in Pybinding. The installation procedure and an example are included in appendix H.

### 3.4 Band structure

The energy band structures for the different TB models with a comparison to the DFT results are presented in figure 3. Although features at the $\Gamma$ and $K$ points in the conduction and valence bands are reasonably well described by most models, only the model from $\text{Liu}_6$ seems to give a good description around the $Q$-point. This model is the only one with a $\delta = 6$ nearest neighbour hopping. This hopping seems to give an important contribution to the shift of the bands at the $Q$-point as can be seen in appendix G. A long range TB model is thus needed to accurately fit the bandstructure at the $Q$-point.

Though the deeper energy bands, far from the band gap are not important in most calculations, they still play a role in determining the orbital contribution of all the other bands. As can be seen in the SK TB models (Rostami, Cappelluti and Dias), these deeper energy bands often have wrong energy values compared to the DFT results. By placing wrong energy values at these deeper energies for the TB model, the model can obtain a better fit for the higher energies. The inexact energy value allows the model to get a correct orbital contribution for bands close to the band gap.

### 3.5 DOS

As can be seen in figure 4 from the DFT result, the orbital contribution around the band gap is mainly given by the $d$-orbitals, $\phi^{M,\rho}$. All TB models have a higher contribution from the $d$-orbitals in the conduction band and the higher valence bands, like as seen in the DFT results. Though the shape of the DOS varies between the models, they agree reasonably well around the valence and conduction band edges.

In our novel TB model and the TB models from $\text{Liu}_2$, $\text{Liu}_6$ and Wu there is a gap in the valence band that mainly originates from the absence of the odd-bands in these models. The DOS for the conduction band from Wu matches the DFT results well, even without the $p$-orbitals.

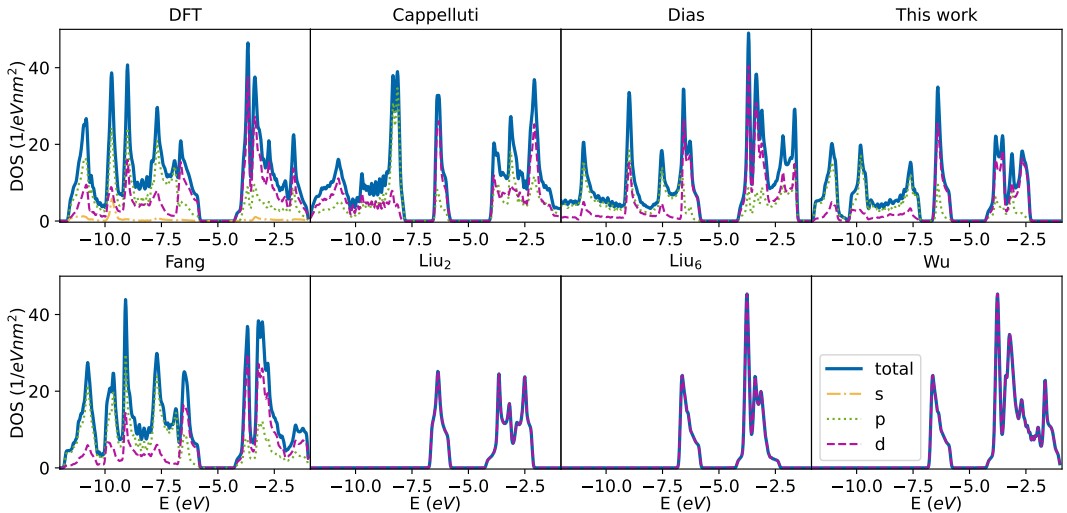

Figure 4: The (projected) DOS for DFT results and the different TB models given in table 1, except for the model from Rostami. The DFT results also include the *s*-orbitals.

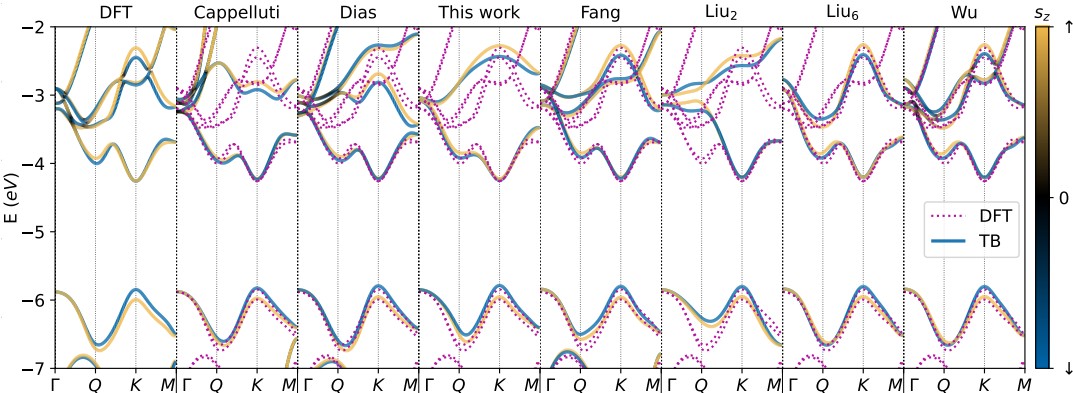

Figure 5: The band structure with SOC, with the projected $s_z$ values for the DFT results and the TB models given in table 1, except for the model from Rostami. Between the $M$ and $\Gamma$ points, the bands are degenerate due to the time reversal symmetry.

In the DOS obtained from the DFT calculations, there are small contributions from the *s*-orbitals. These orbitals are not described by the TB models, they are relatively small and far from the valence and conduction band edges.

## 3.6 SOC

In figure 5 we give the band structure for the TB models where the SOC contribution is included. The orbital contribution determines the splitting of the bands at the $K$-point. It is clear that the inclusion of the small $\mathbf{L} \cdot \mathbf{S}$-term gives good predictions for the splitting and avoided crossings in the band structure. As mentioned in table 1, the spin-flip term is only possible in models that have the orbitals with consecutive $|m_l|$ azimuthal quantum numbers. At the $Q$-point, the local minimum splits, similar to the $K$-point.

The TB models from Liu/Wu describe an effective *d*-orbital, the specific values for the SOC are thus expected to be slightly different than the one from DFT. The metal and chalcogen atoms have different contributions to the $\lambda^\mu$-parameter in equation (18), where as the effective

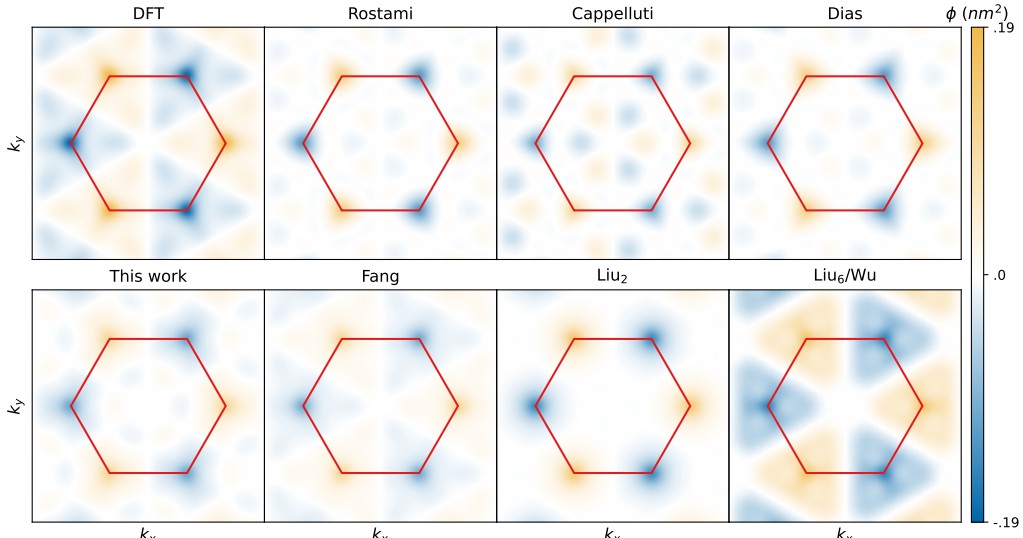

Figure 6: The Berry phase for the different models as given in table 1 and the DFT results. The models from $Liu_6$ and Wu have the same valence band and give the same results. The first Brillouin zone is indicated in red.

orbitals only have one $\lambda^M$-parameter. The Wu TB fitted the spin-degenerate DFT data quite well as seen in figure 3. With the inclusion of SOC, there are small changes as the DFT results gives the same ordering for the energy in the valence band around the $Q$-point as at the $K$-point, but the model from Wu predicts a crossover at the $Q$-point compared with the $K$-point.

## 3.7   Berry phase

The Berry phase is a topological quantity that gives more insight in the accuracy of the wave-functions produced by the TB models. The Berry phase calculation implemented in Pybinding follows the approach introduced in PythTB [36, 37].

The Berry phase obtained from the DFT results looks similar to the one obtained from Fang and $Liu_6$/Wu. The other TB models have local minima/maxima or even a sign change around the $Q$-point that is not present in the DFT results. It seems that the inclusion of the third nearest neighbour hopping improves the results for the Berry phase. On the other hand, the Berry phase in the SK models is less accurate compared to the DFT results than the SG models.

## 4   Discussion

In general, the TB models based on the SK approach give less accurate results compared to the results obtained from DFT than the SG TB models. It appears that the SK approach does not have enough parameters to capture the electronic properties of the TMDs. For the SK-models studied here, only two parameters contribute to the hybridization between the $p$- and $d$-orbitals, $V_{pd\sigma}$ and $V_{pd\pi}$. The limited set of parameters makes these SK TB models sensitive to small changes in these parameters. This makes it hard to find a parameterization that reproduces the specific properties from the DFT results, such as getting a good fit for energy levels around the band gap and the right orbital contributions for these bands.

The even and odd parameterization in Dias creates duplicates for the even and odd sectors. This allows the parameters to describe the finer details in the TB model for the even and odd parts separately. The longer distance hopping in this TB model makes the Hamiltonian overall

denser, which makes it unsuitable for a large scale calculation.

Even when all the different kind of hoppings up to $\delta_{max} = 6$ nearest neighbours are taken into account, we could never obtain a good fit for SK models. If the same long-range model is fitted with a SG TB model, an almost perfect fit with the DFT results is obtained as given in appendix F.

The less accurate results obtained from the SK approach likely originate from the hybridization between the $p$- and $d$-orbitals in the valence band. The SK orbitals are expected to be hydrogen-like atomic orbitals. Instead, SG orbitals can describe the effective orbitals from the hybridization.

This discrepancy between the SK and SG TB models is illustrated below with an example. In the Liu$_2$ SG TB model, only the even $d$-orbitals ($\phi^{M,e}$) are taken into account with only the effective first nearest neighbour hoppings, $\delta = 2M$. In the SK model from Rostami $et\ al$ , this hopping $\delta = 2M$ is also present. Turning off all the interactions between the $p$-orbitals, it results in the same 3-band model.

In appendix E, the Hamiltonian at the $\Gamma$ and $K$ points is described, and the eigenvalues for the TB models from Liu are given. When the SK parameters are included in this description, the following eigenvalues are obtains for the SK and SG approaches at the $\Gamma$-point:

$$
\begin{aligned}
SK \quad \epsilon_{d_0}^{\Gamma}: \quad & \frac{3}{2}V_{dd\sigma}^{2,Me} + \frac{9}{2}V_{dd\delta}^{2,Me} + \Delta_0\,, \\
\epsilon_{d_2}^{\Gamma}: \quad & \frac{9}{4}V_{dd\sigma}^{2,Me} + 3V_{dd\pi}^{2,Me} + \frac{3}{4}V_{dd\delta}^{2,Me} + \Delta_2\,, \\
\epsilon_{d_2}^{\Gamma}: \quad & \frac{9}{4}V_{dd\sigma}^{2,Me} + 3V_{dd\pi}^{2,Me} + \frac{3}{4}V_{dd\delta}^{2,Me} + \Delta_2\,, \\
SG \quad \epsilon_{d_0}^{\Gamma}: \quad & 6u_0^{2,Me} + \varepsilon_0^{M,e}\,, \\
\epsilon_{d_2}^{\Gamma}: \quad & 3u_3^{2,Me} + 3u_5^{2,Me} + \varepsilon_1^{M,e}\,, \\
\epsilon_{d_2}^{\Gamma}: \quad & 3u_3^{2,Me} + 3u_5^{2,Me} + \varepsilon_1^{M,e}\,.
\end{aligned}
\tag{22}
$$

Both the SK and SG TB models lead to similar results, there is a non-degenerate energy level and a twofold degenerate energy level. The value for $\epsilon_{d_0}^{\Gamma}$ is the highest energy level in the valence band, given by the $d_{z^2}$-orbital, the value for $\epsilon_{d_2}^{\Gamma}$ is a degenerate energy level in the conduction band given by a combination of the $\left(d_{x^2-y^2}, d_{xy}\right)$-orbitals.

At the $K$ point, the eigenvalues are

$$
\begin{aligned}
SK \quad \epsilon_{d_0}^{K}: \quad & -\frac{3}{4}V_{dd\sigma}^{2,e} - \frac{9}{4}V_{dd\delta}^{2,e} + \Delta_0\,, \\
\epsilon_{d_{-2}}^{K}: \quad & -\frac{9}{8}V_{dd\sigma}^{2,e} - \frac{3}{2}V_{dd\pi}^{2,e} - \frac{3}{8}V_{dd\delta}^{2,e} + \Delta_2\,, \\
\epsilon_{d_{+2}}^{K}: \quad & -\frac{9}{8}V_{dd\sigma}^{2,e} - \frac{3}{2}V_{dd\pi}^{2,e} - \frac{3}{8}V_{dd\delta}^{2,e} + \Delta_2\,, \\
SG \quad \epsilon_{d_0}^{K}: \quad & -3u_0^{2,Me} + \varepsilon_0^{M,e}\,, \\
\epsilon_{d_{-2}}^{K}: \quad & -\frac{3}{2}u_3^{2,Me} + 3\sqrt{3}u_4^{2,Me} - \frac{3}{2}u_5^{2,Me} + \varepsilon_1^{M,e}\,, \\
\epsilon_{d_{+2}}^{K}: \quad & -\frac{3}{2}u_3^{2,Me} - 3\sqrt{3}u_4^{2,Me} - \frac{3}{2}u_5^{2,Me} + \varepsilon_1^{M,e}\,.
\end{aligned}
\tag{23}
$$

There is a difference between the SK and SG TB models for the values of $\epsilon_{d_{\pm2}}^{K}$. The value of $\epsilon_{d_0}^{K}$ is again given by the $d_{z^2}$-orbital, the values for $\epsilon_{d_{\pm2}}^{K}$ are given by a linear combination of the $\left(d_{x^2-y^2}, d_{xy}\right)$-orbitals, the chiral basis. At the $K$-point the bands are flipped, and the valence bands is given by the linear combination $d_{+2}$ instead of the $d_{z^2}$-orbital. In the SG TB model,

Table 3: Ranking of TB models, which to consider for specific calculations.

|   | Model | Comment |
|---|-------|---------|
| 1 | Liu$_6$ | Low computational cost |
| 2 | This work | Low computational cost, with even $p$-orbitals |
| 3 | Wu | SOC-flip, full $d$-orbitals |
| 4 | Fang | All the orbitals |
| 5 | Rostami | SK-elements needed |
| 6 | Cappelluti | SK-elements needed with all the orbitals |
| 7 | Dias | SK elements needed, more accuracy |
| 8 | Liu$_2$ | When the (effective) 1NN hopping is needed |

the value of $u_4^{2,e}$ breaks the symmetry between the two $d_{\pm 2}$-orbitals. However, in the SK model this parameter is set to zero so that these energy levels are always degenerate at the $K$-point, the SK model can never describe the effective $d$-orbitals.

The inability to construct a SK model with only $d$-orbitals was discussed by Cappelluti *et al* [20]. The model from Liu *et al* [21] was published almost simultaneously. Furthermore, Cappelluti *et al* mentioned that the fit was not good, it only represented the required orbital characteristics for the TB model.

For the 5th nearest neighbour hopping along the armchair direction, the equivalent $u_4^{5,e}$-parameter is zero because of the $\hat{\sigma}_v$-symmetry. The 6th NN hopping goes along the zigzag direction, allowing for another term that breaks the degeneracy of the $d_{\pm 2}$-orbitals at the $K$ point, as mentioned in appendix E. This inclusion of the additional parameters with the 6th NN hopping can contribute to the better description of the local minima in the conduction band at the $Q$-point. Looking at variations for the $u_4^{6M,e}$-parameter indeed reveales that it has a strong influence for the values at the $Q$-point, as discussed in appendix G.

Concluding the previous discussion, we can give a ranking of the different TB models. It is clear that the SK models should be avoided, except when the parameters need to be written in terms of the types of bonds between the orbitals, to describe certain properties like strain [13,38]. The good agreement with DFT and the overall simplicity of Liu$_6$ model means that it is a good start for numerical calculations. Our model also adds the $p$-orbitals, which makes it better suited for investigating small variations in the structure and large scale calculations. The novel model only includes short range hoppings so that the Hamiltonian stays relatively sparse. The model from Fang has all the orbitals, including the full top- and bottom-$p$-orbitals, which makes it useful to investigate heterostructures where there are different hoppings between the top- and bottom-$p$-orbitals. The model from Wu should be used when also the odd $d$-orbitals are needed, for example when the spin-flip needs to be taken into account. The Liu$_2$ model should be used when only the effective first nearest neighbour hopping $\delta_{max} = 1$ is needed. These conclusions are summarized in table 3.

# 5 Conclusion

We gave an overview of the most important TB models from literature, and proposed a novel TB model. The properties of the TB models and the construction method used determines the precision of the description of the electronic structure. While we used MoS$_2$ as an example to study these models, the formalism can be expanded to similar TMDs because of the similar electronic structure. Overall, the TB model from Liu$_6$ gives impressive results around the band gap. The TB model from Fang gives even better overall results, but it has more bands and thus leads to a denser Hamiltonian.

Our newly proposed TB model is more efficient, suitable for large scale calculations where the presence of chalcogen atoms is desired. The models from Liu or Wu do not have the chalcogen atoms in the model, and thus it is hard to model a nanoribbon with specific edges. Our model has the explicit dependence of the chalcogen atoms, while maintaining a minimal amount of hoppings.

Finding a parameterization that matches the symmetry group of the TB model is important. We saw that excluding a specific parameter can lead to a degeneracy that is not expected from the DFT results. This degeneracy is broken if all the parameters allowed by the symmetry group are taken into account. Furthermore, this parameter also helps in obtaining a better fit around the $Q$ point. Our newly proposed model incorporates this additional parameter.

All the models, their new parameters and the parameters from literature are included in a new Python package, TMDybinding [10]. This package uses the same construction method for all the TB models, so that parameters can be compared easily between the different models.

It is clear from this analysis, that there is not one go-to model for $MoS_2$ like there is for graphene. However, one should avoid a TB model that uses the SK approach for any numeric calculations, besides when there is an explicit dependence on the energy for specific type of bonds.

# Acknowledgements

**Funding information** Bert Jorissen acknowledges the support of the Flemish Research Foundation (FWO/11E5821N). The computational resources and services used in this work were provided by the VSC (Flemish Supercomputer Center) and the HPC infrastructure of the University of Antwerp (CalcUA), funded by the Flemish Research Foundation (FWO).

# A   Onsite matrices

The onsite energies are given by

$$
\begin{aligned}
\varepsilon^{X,e} &= \mathrm{diag}\left(\varepsilon_0^{X,e}, \quad \varepsilon_0^{X,e}, \quad \varepsilon_1^{X,e}\right), & \varepsilon^{X,o} &= \mathrm{diag}\left(\varepsilon_0^{X,o}, \quad \varepsilon_0^{X,o}, \quad \varepsilon_1^{X,o}\right), \\
\varepsilon^{M,e} &= \mathrm{diag}\left(\varepsilon_0^{M,e}, \quad \varepsilon_1^{M,e}, \quad \varepsilon_1^{M,e}\right), & \varepsilon^{M,o} &= \mathrm{diag}\left(\varepsilon_0^{M,o}, \quad \varepsilon_0^{M,o}\right).
\end{aligned}
\tag{A.1}
$$



Table 4: The fitted parameters for the SG TB models, as used in this manuscript.

| Param | Liu$_2$ | Liu$_6$ | Wu | This work | Fang | All |
|---|---|---|---|---|---|---|
| $\epsilon_0^{M,e}$ | -4.752 | -5.098 | -5.098 | -6.475 | -6.486 | -4.110 |
| $\epsilon_1^{M,e}$ | -3.812 | -4.101 | -4.101 | -4.891 | -5.185 | -3.508 |
| $\epsilon_0^{M,o}$ | | | -2.246 | | -4.594 | -2.812 |
| $\epsilon_0^{X,e}$ | | | | -7.907 | -7.758 | -5.855 |
| $\epsilon_1^{X,e}$ | | | | -9.470 | -9.051 | -6.550 |
| $\epsilon_0^{X,o}$ | | | | | -7.233 | -4.965 |
| $\epsilon_1^{X,o}$ | | | | | -7.027 | -4.733 |
| $u_0^{1,e}$ | | | | 0.999 | 1.268 | 1.358 |
| $u_1^{1,e}$ | | | | -1.289 | -1.087 | -0.940 |
| $u_2^{1,e}$ | | | | 0.795 | 0.731 | 0.636 |
| $u_3^{1,e}$ | | | | -0.688 | -0.690 | -0.866 |
| $u_4^{1,e}$ | | | | -0.795 | -0.845 | -0.941 |
| $u_0^{1,o}$ | | | | | -0.782 | -0.782 |
| $u_1^{1,o}$ | | | | | 2.195 | 2.138 |
| $u_2^{1,o}$ | | | | | -1.317 | -1.455 |
| $u_0^{2,Me}$ | -0.183 | -0.143 | -0.143 | -0.048 | -0.035 | -0.289 |
| $u_1^{2,Me}$ | 0.560 | 0.509 | 0.509 | 0.580 | 0.473 | 0.508 |
| $u_2^{2,Me}$ | 0.350 | -0.114 | -0.114 | -0.074 | 0.124 | 0.118 |
| $u_3^{2,Me}$ | 0.026 | 0.080 | 0.079 | -0.414 | -0.397 | -0.554 |
| $u_4^{2,Me}$ | -0.325 | -0.163 | -0.164 | -0.299 | -0.259 | -0.229 |
| $u_5^{2,Me}$ | 0.222 | 0.085 | 0.085 | 0.045 | 0.147 | 0.230 |
| $u_0^{2,Mo}$ | | | -0.167 | | -0.164 | -0.262 |
| $u_1^{2,Mo}$ | | | 0.022 | | -0.106 | 0.216 |
| $u_2^{2,Mo}$ | | | -0.148 | | -0.089 | 0.010 |
| $u_0^{2,Xe}$ | | | | 0.795 | 0.872 | 0.911 |
| $u_1^{2,Xe}$ | | | | -0.248 | -0.133 | -0.022 |
| $u_2^{2,Xe}$ | | | | 0.164 | 0.067 | 0.096 |
| $u_3^{2,Xe}$ | | | | -0.002 | -0.087 | 0.003 |
| $u_4^{2,Xe}$ | | | | -0.293 | -0.167 | -0.028 |
| $u_5^{2,Xe}$ | | | | -0.174 | -0.256 | -0.184 |
| $u_0^{2,Xo}$ | | | | | 0.779 | 0.850 |
| $u_1^{2,Xo}$ | | | | | -0.056 | 0.071 |
| $u_2^{2,Xo}$ | | | | | -0.055 | -0.041 |
| $u_3^{2,Xo}$ | | | | | -0.076 | -0.113 |
| $u_4^{2,Xo}$ | | | | | 0.013 | 0.041 |
| $u_5^{2,Xo}$ | | | | | -0.098 | -0.173 |

Table 5: The fitted parameters for the SG TB models, as used in this manuscript.

| Param | Fang | All |
|---|---|---|
| $u_0^{3,e}$ | | 0.082 |
| $u_1^{3,e}$ | -0.309 | -0.137 |
| $u_2^{3,e}$ | -0.144 | -0.232 |
| $u_3^{3,e}$ | 0.020 | -0.076 |
| $u_4^{3,e}$ | -0.371 | -0.218 |
| $u_0^{3,o}$ | | 0.031 |
| $u_1^{3,o}$ | | -0.064 |
| $u_2^{3,o}$ | | 0.058 |
| $u_0^{4,e}$ | | -0.011 |
| $u_1^{4,e}$ | | 0.001 |
| $u_2^{4,e}$ | | -0.032 |
| $u_3^{4,e}$ | | 0.001 |
| $u_4^{4,e}$ | | 0.029 |
| $u_0^{4,o}$ | | -0.034 |
| $u_1^{4,o}$ | | 0.063 |
| $u_2^{4,o}$ | | 0.051 |

| Param | Liu$_6$ | Wu | All |
|---|---|---|---|
| $u_0^{5,Me}$ | 0.058 | 0.058 | 0.001 |
| $u_1^{5,Me}$ | -0.074 | -0.074 | -0.008 |
| $u_3^{5,Me}$ | -0.040 | -0.040 | 0.034 |
| $u_5^{5,Me}$ | 0.180 | 0.179 | 0.009 |
| $u_6^{5,Me}$ | 0.265 | 0.265 | -0.005 |
| $u_0^{5,Mo}$ | | -0.061 | -0.065 |
| $u_2^{5,Mo}$ | | 0.037 | 0.129 |
| $u_0^{5,Xe}$ | | | -0.013 |
| $u_2^{5,Xe}$ | | | -0.000 |
| $u_3^{5,Xe}$ | | | 0.002 |
| $u_5^{5,Xe}$ | | | 0.003 |
| $u_6^{5,Xe}$ | | | -0.005 |
| $u_0^{5,Xo}$ | | | -0.027 |
| $u_2^{5,Xo}$ | | | -0.008 |
| $u_3^{5,Xo}$ | | | -0.072 |
| $u_5^{5,Xo}$ | | | 0.010 |
| $u_6^{5,Xo}$ | | | -0.024 |
| $u_0^{6,Me}$ | -0.038 | -0.038 | 0.000 |
| $u_1^{6,Me}$ | 0.004 | 0.004 | -0.029 |
| $u_2^{6,Me}$ | 0.045 | 0.045 | 0.000 |
| $u_3^{6,Me}$ | -0.155 | -0.155 | -0.056 |
| $u_4^{6,Me}$ | 0.177 | 0.177 | 0.061 |
| $u_5^{6,Me}$ | 0.270 | 0.270 | 0.073 |
| $u_0^{6,Mo}$ | | 0.169 | 0.043 |
| $u_1^{6,Mo}$ | | -0.149 | -0.060 |
| $u_2^{6,Mo}$ | | -0.123 | -0.127 |
| $u_0^{6,Xe}$ | | | 0.049 |
| $u_1^{6,Xe}$ | | | 0.070 |
| $u_2^{6,Xe}$ | | | 0.007 |
| $u_3^{6,Xe}$ | | | 0.030 |
| $u_4^{6,Xe}$ | | | 0.069 |
| $u_5^{6,Xe}$ | | | -0.002 |
| $u_0^{6,Xo}$ | | | 0.073 |
| $u_1^{6,Xo}$ | | | 0.044 |
| $u_2^{6,Xo}$ | | | 0.009 |
| $u_3^{6,Xo}$ | | | 0.029 |
| $u_4^{6,Xo}$ | | | -0.029 |
| $u_5^{6,Xo}$ | | | 0.023 |

# B Hopping matrices

The hopping matrices are given by

$$
\begin{aligned}
T_1^{n,e} &= \begin{pmatrix} 0 & 0 & u_0^{n,e} \\ u_1^{n,e} & u_2^{n,e} & 0 \\ u_3^{n,e} & u_4^{n,e} & 0 \end{pmatrix}, & T_1^{n,o} &= \begin{pmatrix} u_0^{n,o} & 0 \\ 0 & u_1^{n,o} \\ 0 & u_2^{n,o} \end{pmatrix}, \\
T_1^{m,X\rho} &= \begin{pmatrix} u_0^{m,X\rho} & u_1^{m,X\rho} & u_2^{m,X\rho} \\ -u_1^{m,X\rho} & u_3^{m,X\rho} & u_4^{m,X\rho} \\ -u_2^{m,X\rho} & u_4^{m,X\rho} & u_5^{m,X\rho} \end{pmatrix}, & T_1^{m,Me} &= \begin{pmatrix} u_0^{m,Me} & u_1^{m,Me} & u_2^{m,Me} \\ u_1^{m,Me} & u_3^{m,Me} & u_4^{m,Me} \\ -u_2^{m,Me} & -u_4^{m,Me} & u_5^{m,Me} \end{pmatrix}, \\
T_1^{m,Mo} &= \begin{pmatrix} u_0^{m,Mo} & u_1^{m,Mo} \\ -u_1^{m,Mo} & u_2^{m,Mo} \end{pmatrix}, & T_1^{5,X\rho} &= \begin{pmatrix} u_3^{5,X\rho} & 0 & 0 \\ 0 & u_0^{5,X\rho} & u_2^{5,X\rho} \\ 0 & u_6^{5,X\rho} & u_5^{5,X\rho} \end{pmatrix}, \\
T_1^{5,Me} &= \begin{pmatrix} u_0^{5,Me} & -u_1^{5,Me} & 0 \\ -u_6^{5,Me} & u_3^{5,Me} & 0 \\ 0 & 0 & u_5^{5,Me} \end{pmatrix}, & T_1^{5,Mo} &= \begin{pmatrix} u_2^{5,Mo} & 0 \\ 0 & u_0^{5,Mo} \end{pmatrix},
\end{aligned}
\tag{B.1}
$$

with $m = (2/6)$, $f = (2/5/6)$ and $n = (1/3/4)$. For the 4-th NN hopping, the matrices must also be rotated to obtain

$$
T_1^{4a,\mu\rho} = R^{\mu,\rho} T_1^{4,\mu\rho} (R^{\mu,\rho})^T, \qquad T_1^{4b,\mu\rho} = (R^{\mu,\rho})^T T_1^{4,\mu\rho} R^{\mu,\rho},
\tag{B.2}
$$

with $\gamma = \arctan\left(\frac{\sqrt{3}}{5}\right)$.

The parameters for the hopping matrices for the SG TB models are given in tables 4 and 5.

# C SOC matrices

The matrices for the SOC are given by

$$
t_{SOC}^{\uparrow\uparrow,ee} = \begin{pmatrix} 0 & 0 & 0 & 0 & 0 & 0 \\ 0 & 0 & -i\lambda_M & 0 & 0 & 0 \\ 0 & i\lambda_M & 0 & 0 & 0 & 0 \\ 0 & 0 & 0 & 0 & -i\frac{1}{2}\lambda_X & 0 \\ 0 & 0 & 0 & i\frac{1}{2}\lambda_X & 0 & 0 \\ 0 & 0 & 0 & 0 & 0 & 0 \end{pmatrix},
$$

$$
T_{SOC}^{\uparrow\uparrow,oo} = \frac{1}{2}\begin{pmatrix} 0 & -i\lambda_M & 0 & 0 & 0 \\ i\lambda_M & 0 & 0 & 0 & 0 \\ 0 & 0 & 0 & -i\lambda_X & 0 \\ 0 & 0 & i\lambda_X & 0 & 0 \\ 0 & 0 & 0 & 0 & 0 \end{pmatrix},
$$

$$
T_{SOC}^{\uparrow\downarrow,eo} = \frac{1}{2}\begin{pmatrix} -\sqrt{3}\lambda_M & i\sqrt{3}\lambda_M & 0 & 0 & 0 \\ \lambda_M & i\lambda_M & 0 & 0 & 0 \\ -i\lambda_M & \lambda_M & 0 & 0 & 0 \\ 0 & 0 & 0 & 0 & \lambda_X \\ 0 & 0 & 0 & 0 & -i\lambda_X \\ 0 & 0 & -\lambda_X & i\lambda_X & 0 \end{pmatrix},
$$

$$
T_{SOC}^{\downarrow\uparrow,eo} = \frac{1}{2}\begin{pmatrix} \sqrt{3}\lambda_M & i\sqrt{3}\lambda_M & 0 & 0 & 0 \\ -\lambda_M & i\lambda_M & 0 & 0 & 0 \\ -i\lambda_M & -\lambda_M & 0 & 0 & 0 \\ 0 & 0 & 0 & 0 & -\lambda_X \\ 0 & 0 & 0 & 0 & -i\lambda_X \\ 0 & 0 & \lambda_X & i\lambda_X & 0 \end{pmatrix},
$$

(C.1)

with the following relations between the different parts

$$
T_{SOC}^{\uparrow\uparrow,ee} = -t_{SOC}^{\downarrow\downarrow,ee}, \qquad T_{SOC}^{\uparrow\uparrow,oo} = -t_{SOC}^{\downarrow\downarrow,oo}, \qquad T_{SOC}^{\uparrow\downarrow,eo} = \left(t_{SOC}^{\downarrow\uparrow,oe}\right)^{\dagger}, \qquad T_{SOC}^{\downarrow\uparrow,eo} = \left(t_{SOC}^{\downarrow\uparrow,oe}\right)^{\dagger}. \quad (C.2)
$$

In this work, we fitted the SOC parameters to the DFT results, starting from the spin-degenerate parameterization from the all band model in appendix F, which gave the parameters (in $eV/\hbar^2$)

$$
\lambda^M = 0.0851, \qquad \lambda^X = 0.0673. \quad (C.3)
$$

Table 6: The fitted parameters (in $eV$) for the SK TB models, as used in this manuscript. The positions with a dash ($-$) have parameters that have the same value as the one with the different parity $\rho$.

| Params | Rostami | Cappelluti | Dias | Params | Dias |
|---|---|---|---|---|---|
| $\Delta_0$ | -6.254 | -6.254 | -6.325 | $V_{dd\sigma}^{5,Me}$ | 0.032 |
| $\Delta_1$ | | -7.667 | -4.777 | $V_{dd\pi}^{5,Me}$ | 0.057 |
| $\Delta_2$ | -5.642 | -5.642 | -5.902 | $V_{dd\delta}^{5,Me}$ | -0.038 |
| $\Delta_p$ | -10.248 | -10.248 | -9.523 | $V_{pp\sigma}^{5,Xe}$ | 0.161 |
| $\Delta_z$ | -16.617 | -9.108 | -12.503 | $V_{pp\pi}^{5,Xe}$ | 0.002 |
| $V_{pp\sigma}^0$ | | 7.509 | 5.630 | $V_{dd\pi}^{5,Mo}$ | 0.001 |
| $V_{pp\pi}^0$ | | -0.629 | -2.447 | $V_{dd\delta}^{5,Mo}$ | 0.066 |
| $V_{pd\sigma}^{1,e}$ | 3.892 | 3.892 | 4.009 | $V_{pp\sigma}^{5,Xo}$ | -0.034 |
| $V_{pd\pi}^{1,e}$ | -1.365 | -1.365 | -1.595 | $V_{pp\pi}^{5,Xo}$ | -0.022 |
| $V_{pd\sigma}^{1,o}$ | — | — | 2.158 | | |
| $V_{pd\pi}^{1,o}$ | — | — | -1.069 | | |
| $V_{dd\sigma}^{2,Me}$ | -0.733 | -0.733 | -0.658 | | |
| $V_{dd\pi}^{2,Me}$ | 0.655 | 0.655 | 0.572 | | |
| $V_{dd\delta}^{2,Me}$ | 0.262 | 0.262 | 0.262 | | |
| $V_{pp\sigma}^{2,Xe}$ | 0.734 | 0.734 | 0.993 | | |
| $V_{pp\pi}^{2,Xe}$ | -1.449 | -1.449 | -1.668 | | |
| $V_{dd\pi}^{2,Mo}$ | — | — | 0.043 | | |
| $V_{dd\delta}^{2,Mo}$ | — | — | 0.014 | | |
| $V_{pp\sigma}^{2,Xo}$ | — | — | 0.574 | | |
| $V_{pp\pi}^{2,Xo}$ | — | — | -0.154 | | |

# D  Slater-Koster parameterization

For the Slater-Koster models, the values of the parameters are, with $c = \tan\theta$

$$\varepsilon_0^{X,e} = \Delta_p + V_{pp\pi}^0, \qquad \varepsilon_1^{X,e} = \Delta_z - V_{pp\sigma}^0, \qquad \varepsilon_0^{X,o} = \Delta_p - V_{pp\pi}^0, \qquad \varepsilon_1^{X,o} = \Delta_z + V_{pp\sigma}^0$$
$$\varepsilon_0^{M,e} = \Delta_0, \qquad \varepsilon_1^{M,e} = \Delta_2, \qquad \varepsilon_0^{M,o} = \Delta_1,$$

$$\text{(D.1)}$$

$$u_0^{n,e} = \frac{\sqrt{2} r_n V_{pd\pi}^{n,e}}{\sqrt{r_n^2 + c^2}},$$

$$u_1^{n,e} = -r_n \frac{2\sqrt{3} c^2 V_{pd\pi}^{n,e} + \left(r_n^2 - 2c^2\right) V_{pd\sigma}^{n,e}}{\sqrt{2}\left(r_n^2 + c^2\right)^{\frac{3}{2}}}, \qquad u_2^{n,e} = -r_n \frac{2c^2 V_{pd\pi}^{n,e} + \sqrt{3} r_n^2 V_{pd\sigma}^{n,e}}{\sqrt{2}\left(r_n^2 + c^2\right)^{\frac{3}{2}}},$$

$$u_3^{n,e} = c \frac{r_n^2 2\sqrt{3} c^2 V_{pd\pi}^{n,e} + \left(2c^2 - r_n^2\right) V_{pd\sigma}^{n,e}}{\sqrt{2}\left(r_n^2 + c^2\right)^{\frac{3}{2}}}, \qquad u_4^{n,e} = r_n^2 c \frac{2 V_{pd\pi}^{n,e} - \sqrt{3} V_{pd\sigma}^{n,e}}{\sqrt{2}\left(r_n^2 + c^2\right)^{\frac{3}{2}}},$$

$$u_0^{n,o} = \frac{\sqrt{2} c V_{pd\pi}^{n,o}}{\sqrt{r_n^2 + c^2}},$$

$$u_1^{n,o} = \sqrt{2} c \frac{\left(c^2 - r_n^2\right) V_{pd\pi}^{n,o} + r_n^2 \sqrt{3} V_{pd\sigma}^{n,o}}{\sqrt{2}\left(r_n^2 + c^2\right)^{\frac{3}{2}}}, \qquad u_2^{n,o} = \sqrt{2} r_n \frac{\left(r_n^2 - c^2\right) V_{pd\pi}^{n,o} + c^2 \sqrt{3} V_{pd\sigma}^{n,o}}{\sqrt{2}\left(r_n^2 + c^2\right)^{\frac{3}{2}}},$$

$$\text{(D.2)}$$

$$u_0^{f,Xe} = V_{pp\sigma}^{f,Xe} + V_{pp\pi,tb}^{f,Xe} - r_f^2 \frac{V_{pp\pi,tb}^{f,Xe} - V_{pp\sigma,tb}^{f,Xe}}{r_f^2 + 4c^2}, \qquad u_1^{m,Xe} = 0,$$

$$u_2^{f,Xe} = -2c r_f \frac{V_{pp\pi,tb}^{f,Xe} - V_{pp\sigma,tb}^{f,Xe}}{r_f^2 + 4c^2}, \qquad u_3^{f,Xe} = V_{pp\pi}^{f,Xe} + V_{pp\pi,tb}^{f,Xe}, \qquad u_4^{m,Xe} = 0,$$

$$u_5^{f,Xe} = V_{pp\pi}^{f,Xe} - V_{pp\sigma,tb}^{f,Xe} - r_f^2 \frac{V_{pp\pi,tb}^{f,Xe} - V_{pp\sigma,tb}^{f,Xe}}{r_f^2 + 4c^2}, \qquad u_6^{5,X\rho} = -u_2^{5,X\rho}$$

$$u_0^{f,Xo} = V_{pp\sigma}^{f,Xo} - V_{pp\pi,tb}^{f,Xo} + r_f^2 \frac{V_{pp\pi,tb}^{f,Xo} - V_{pp\sigma,tb}^{f,Xo}}{r_f^2 + 4c^2}, \qquad u_1^{m,Xo} = 0,$$

$$u_2^{f,Xo} = 2c r_f \frac{V_{pp\pi,tb}^{f,Xo} - V_{pp\sigma,tb}^{f,Xo}}{r_f^2 + 4c^2}, \qquad u_3^{f,Xo} = V_{pp\pi}^{f,Xo} - V_{pp\pi,tb}^{f,Xo}, \qquad u_4^{m,Xo} = 0,$$

$$u_5^{f,Xo} = V_{pp\pi}^{f,Xo} + V_{pp\sigma,tb}^{f,Xo} + r_f^2 \frac{V_{pp\pi,tb}^{f,Xo} - V_{pp\sigma,tb}^{f,Xo}}{r_f^2 + 4c^2},$$

$$\text{(D.3)}$$

$$u_0^{f,Me} = \frac{1}{4}\left(V_{dd\sigma}^{f,Me} + 3 V_{dd\delta}^{f,Me}\right), \qquad u_1^{f,Me} = \frac{\sqrt{3}}{4}\left(-V_{dd\sigma}^{f,Me} + V_{dd\delta}^{f,Me}\right), \qquad u_2^{m,Me} = 0,$$

$$u_3^{f,Me} = \frac{1}{4}\left(3 V_{dd\sigma}^{f,Me} + V_{dd\delta}^{f,Me}\right), \qquad u_4^{m,Me} = 0, \qquad u_5^{f,Me} = V_{dd\pi}^{f,Me},$$

$$u_6^{5,Me} = u_1^{5,Me}$$

$$u_0^{f,Mo} = V_{dd\pi}^{f,Mo}, \qquad u_1^{m,Mo} = 0, \qquad u_2^{f,Mo} = V_{dd\delta}^{f,Mo},$$

$$\text{(D.4)}$$

with $r_1 = -1$, $r_2 = \sqrt{3}$, $r_3 = 2$, $r_4 = -\sqrt{7}$, $r_5 = 3$ and $r_6 = 2\sqrt{3}$. The parameters for the hopping matrices for the SK TB models are given in table 6.

# E  Hamiltonian at high-symmetry points

At the high-symmetry points $\Gamma$ and $K$, the Hamiltonian can be solved exactly by preforming suitable basis transformations [20].

## E.1  $\Gamma$-point

At the $\Gamma$-point, the Hamiltonian becomes block-diagonal in the following Cartesian basis

$$\phi = \left( \phi^e_{p_z d_0}, \phi^e_{p_x d_2}, \phi^e_{p_y d_2}, \phi^o_{p_x d_1}, \phi^o_{p_y d_1}, \phi^o_{p_z} \right), \tag{E.1}$$

with

$$
\begin{aligned}
\phi^e_{p_z d_0} &= \left( p^e_z, d_{z^2} \right), & \phi^e_{p_x d_2} &= \left( p^e_x, d_{xy} \right), & \phi^e_{p_y d_2} &= \left( p^e_y, d_{x^2-y^2} \right), \\
\phi^o_{p_x d_1} &= \left( p^o_x, d_{xz} \right), & \phi^o_{p_y d_1} &= \left( p^o_y, d_{yz} \right), & \phi^o_{p_z} &= \left( p^o_z \right),
\end{aligned}
\tag{E.2}
$$

making $2 \times 2$ blocks for all the groups of orbitals, except for the $\phi^o_{p_z}$ which is a $1 \times 1$ block with only the $p^o_z$-orbital.

Each block has contributions for a metal $d$ and a chalcogen $p$ orbital. This allows us to denote these blocks as

$$H^\Gamma_{pd} = \begin{pmatrix} u_d & u_{pd} \\ u^\dagger_{pd} & u_p \end{pmatrix}. \tag{E.3}$$

The eigenvalues of such a block are given by

$$\epsilon^\Gamma_{pd} = \frac{1}{2} \left( u_d + u_p \pm \sqrt{\left( u_d - u_p \right)^2 + 4|u_{pd}|^2} \right). \tag{E.4}$$

The terms on the diagonal $u_p$ and $u_d$, get contributions from the onsite energies and the second, fifth and sixth nearest neighbour hoppings (given by $f$), the $pd$ terms only get contributions from the first, third and fourth nearest neighbour hoppings (given by $n$).

In the 3-band TB model with only even $d$-orbitals, the second fifth and sixth nearest neighbour hoppings are present. This results in $u_{pd}$ being zero. In this model, there are no $p$-orbitals, the $u_p$ disappears and the Hamiltonian becomes diagonal. The eigenvalues are easily obtained as

$$
\begin{aligned}
\epsilon^\Gamma_{d_0} &= \varepsilon^{M,e}_0 + 6 \left( u^{2,Me}_0 + u^{5,Me}_0 + u^{6,Me}_0 \right), \\
\epsilon^\Gamma_{d_2} &= \varepsilon^{M,e}_1 + 3 \left( u^{2,Me}_3 + u^{5,Me}_3 + u^{6,Me}_3 + u^{2,Me}_5 + u^{5,Me}_5 + u^{6,Me}_5 \right),
\end{aligned}
\tag{E.5}
$$

where the level $\varepsilon_{d_0}$ originates from the $d_{z^2}$-orbital, and the $\varepsilon_{d_2}$ is degenerate between the $d_{xy}$- and $d_{x^2-y^2}$-orbitals.

## E.2  $K$-point

At the $K$-point, the Hamiltonian becomes block-diagonal in the following chiral basis:

$$\phi = \left( \phi^e_{p^e_{-1} d_{-2}}, \phi^e_{p^e_0 d_{+2}}, \phi^e_{p^e_{+1} d_0}, \phi^o_{p^o_{-1} d_{+1}}, \phi^o_{p^o_0 d_{-1}}, \phi^o_{p^o_{+1}} \right), \tag{E.6}$$

with

$$\phi^e_{p^e_{-1}d_{-2}} = \left(\frac{1}{\sqrt{2}}p^e_x - \frac{i}{\sqrt{2}}p^e_y, \frac{1}{\sqrt{2}}d_{x^2-y^2} - \frac{i}{\sqrt{2}}d_{xy}\right), \qquad \phi^e_{p^e_0 d_{+2}} = \left(p^e_z, \frac{1}{\sqrt{2}}d_{x^2-y^2} + \frac{i}{\sqrt{2}}d_{xy}\right),$$

$$\phi^o_{p^o_{-1}d_{+1}} = \left(\frac{1}{\sqrt{2}}p^o_x - \frac{i}{\sqrt{2}}p^o_y, \frac{-1}{\sqrt{2}}d_{xz} - \frac{i}{\sqrt{2}}d_{yz}\right), \qquad \phi^e_{p^e_{+1}d_0} = \left(\frac{-1}{\sqrt{2}}p^e_x - \frac{i}{\sqrt{2}}p^e_y, d_{z^2}\right),$$

$$\phi^o_{p^o_0 d_{-1}} = \left(p^o_z, \frac{1}{\sqrt{2}}d_{xz} - \frac{i}{\sqrt{2}}d_{yz}\right), \qquad \phi^o_{p^o_{+1}} = \left(\frac{-1}{\sqrt{2}}p^o_x - \frac{i}{\sqrt{2}}p^o_y\right),$$

$$\tag{E.7}$$

again making $2 \times 2$ blocks and one $1 \times 1$ block, like explained for the case at the $\Gamma$-point. The eigenvalues can also be obtained in the same manner. The same hoppings still contribute to the diagonal or off-diagonal terms.

For the 3-band TB model the Hamiltonian again becomes diagonal, the eigenvalues are given by

$$\epsilon^K_{d_0} = \varepsilon^{M,e}_0 - 3\left(u^{2,Me}_0 - 2u^{5,Me}_0 + u^{6,Me}_0\right),$$

$$\epsilon^K_{d_{-2}} = \varepsilon^{M,e}_1 - \frac{3}{2}\left(u^{2,Me}_3 - 2u^{5,Me}_3 + u^{6,Me}_3 + u^{2,Me}_5 - 2u^{5,Me}_5 + u^{6,Me}_5\right)$$
$$+ 3\sqrt{3}\left(u^{2,Me}_4 - u^{6,Me}_4\right),$$

$$\epsilon^K_{d_{+2}} = \varepsilon^{M,e}_1 - \frac{3}{2}\left(u^{2,Me}_3 - 2u^{5,Me}_3 + u^{6,Me}_3 + u^{2,Me}_5 - 2u^{5,Me}_5 + u^{6,Me}_5\right)$$
$$- 3\sqrt{3}\left(u^{2,Me}_4 - u^{6,Me}_4\right),$$

$$\tag{E.8}$$

giving three distinct energy levels. The contributions to the last two energy levels are given by the $d_2$-states, thus resulting in a superposition of $d_{x^2-y^2}$ and $d_{xy}$.

In theory, one can construct a TB model consisting of only two bands: a $d_{z^2}$ and $d_{+2}$-band. However, this will only be correct for the energy levels close to the band gap. It would model an effective $d$-orbital for the conduction band, thus resulting in an effective TB model of effective $d$-orbitals.

The $d$-orbitals are degenerate if the $u^{m,Me}_4$-orbitals are zero, the hoppings along the zigzag direction. The fifth nearest neighbour hopping, along the armchair direction, does not contribute to this splitting because it is protected by the $\sigma_v$-symmetry.

# F All hoppings

To further analyze all the results, we did a fit up to sixth nearest neighbours. This included all the hoppings for all the orbitals. The fit corresponds quite well with the DFT results, as seen in figure 7. The discrepancies of the energy levels align with the energy levels where the $s$-orbital has a higher orbital contribution, as seen in the DFT results for the DOS in figure 4. The parameters for this SG TB model are given in table 4 and 5.

# G Sensitivity of the $Q$-point

The $Q$-point is sensitive to longer range hoppings. From figure 3 we can see that the $Q$-point has a better agreement with the DFT results for TB models that include higher order hoppings. To quantify this better, we investigated the sensitivity of the $\text{Liu}_6$ SG TB model to its parameters. As the case with the band gap opening up, the $u^4_6$-parameter seems to play a role. However, this parameter also shifts the location of the location of the energy of the valence band. This

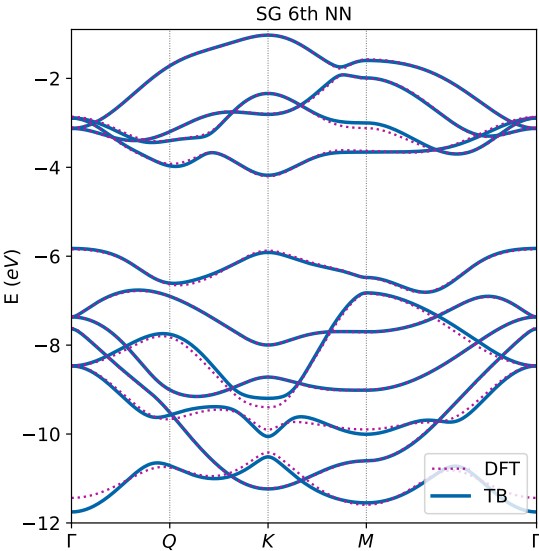

Figure 7: A SG TB model for all the orbitals and hoppings up to sixth nearest neighbours.

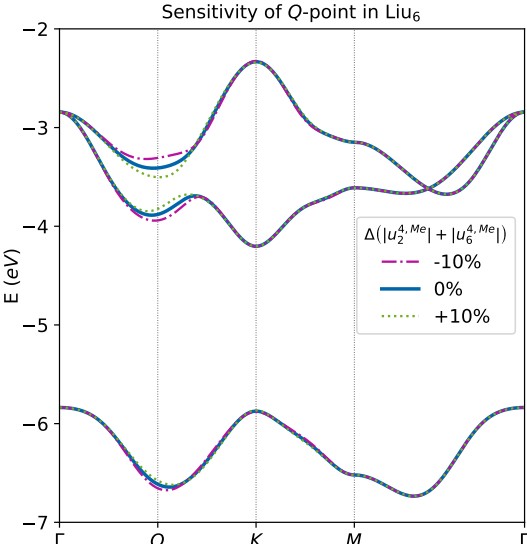

Figure 8: Sensitivity of the $Liu_6$ SG TB model to changes in the parameters $u_2^{4,Me}$ and $u_4^{6,Me}$. If both values are scaled so that they compensate each other, only the energy levels around the $Q$-point change.

shift can be compensated by changing the $u_2^4$-parameter. We scale the parameters so with a percentage $\Delta$ given as

$$u_2^{4Me} \rightarrow u_2^{4Me} + \Delta|u_2^{4Me}|, \qquad u_6^{4Me} \rightarrow u_6^{4Me} + \Delta|u_6^{4Me}|. \tag{G.1}$$

This compensated shift is given in figure 8. In this figure, both parameters are scaled by ±10%.

## H TMDybinding

We present a new code, TMDybinding [10], that allows to easily calculate the TB models for TMDs with pybinding. In this code, the symmetries of the TMDs are taken into account automatically. Only the values for a few (SK) parameters need to be given, the other parameters are computed in the background.

The installation is simple, just run:

```
pip install tmdybinding
```

As an example, we calculate the band structure for the Liu$_6$ TB model along the path through $\Gamma$, $K$, $M$ and $\Gamma$, as given in figure 8 for the case of $\Delta = 0\%$,

```
import pybinding as pb
import tmdybinding as td
from tmdybinding.sg_parameters import liu6

# construct the unit cell
lat = td.TmdNN256Meo(params=liu6["MoS2"]).lattice()
# make a periodic system
model = pb.Model(lat, pb.translational_symmetry())
# get the corners from the Brillouin zone
bz = lat.brillouin_zone()
# calculate along the high symmetry path
bands = pb.solver.lapack(model).calc_bands(
    bz[3] * 0, bz[3], (bz[3] + bz[4]) / 2, bz[3] * 0
)
# visualize the results
bands.plot(point_labels=[
    r"$\Gamma$", "$K$", "$M$", r"$\Gamma$"
])
```

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
