# Peer review of "Comparative Analysis of Tight-Binding models for Transition Metal Dichalcogenides"

_SciPost Physics Core, doi:SciPost Phys. Core 7, 004 (2024)_

## Round 1 · Referee Report · Anonymous (Referee 1) · 2023-12-15

Report

In this paper the Authors provide a detailed overview of different tight-binding (TB) model available in literature for transition-metal dichalcogenides (TMDs), discussing pros and cons of each model.
They also present a their own TB model suitable for computation of large-scale systems.
Furthermore, the Authors make available for the scientific community a Phyton-based code for all the TB models.

I deeply thank the Authors for this work.
The theoretical discussion is clear, detailed, useful and exhaustive, and the public code is a profound service for the community.

Of course, I warmly recommend publication.

---

## Round 1 · Referee Report · Anonymous (Referee 2) · 2024-1-5

Strengths

Given the rapid evolution of research in TMD materials and the multitude of publications utilizing various lattice models for electronic structures, this paper fills a significant gap in the field by:

  1. Providing a clear comparative overview of different tight-binding models and discussing their validity range.
  2. Offering basic instructions on developing TMD tight-binding models, particularly beneficial for non-experts in the field.
  3. It propose a new computational efficient tight-binding model
  4. Supplying Python code for the implementation of multi-band tight-binding models.

Weaknesses

  1. Presentation issue: The paper lacks a distinct section heading that addresses the new tight-binding model developed in this study.

  2. Citation issue: For completeness, the authors may consider including all original papers on the tight-binding model of TMDs.

Report

This paper provides a clear and useful overview of tight-binding models for TMDs, and I believe it can be beneficial for researchers entering this field. Specifically, this work can facilitate the study of heterostructures involving TMD materials. In general, I find this work valuable with satisfactory quality and believe it is suitable for publication in Scipost.

Requested changes

It would be beneficial to include a section heading that clearly presents the new developments presented in this paper. The author can improve the bibliography section in order to make it more complete as an overview to TMD lattice models. For example, they may consider the following papers to add to their bibliography section:

F. Zahid et al, AIP Adv. 3, 052111 (2013)
H. Rostami et al, Phys. Rev. B 88, 085440 (2013)

---

## Round 2 · Referee Report · Anonymous (Referee 2) · 2024-1-20

Report

The revised version has thoroughly addressed all previously raised concerns. Based on the improvements made, I highly recommend this work for publication.

---

## Round 2 · List of Changes

• Added references for a complete list of important tight-binding models in literature for the TMDs studied in the manuscript
  • Added a new subsection title and elaborated more on the specific details of the novel tight-binding model
  • Some minor typos and adjustments

---

## Editorial Decision

published